# Study on Soil Total Nitrogen Content Prediction Method Based on Synthetic Neural Network Model

He Liu [1], Jiamu Wang [1], Shuyan Liu [2], Qingran Hu [1] and Dongyan Huang [3,*]

[1] College of Information Technology, Jilin Agricultural University, Changchun 130118, China; liuhe@jlau.edu.cn (H.L.); 20221124@mails.jlau.edu.cn (J.W.); huqr20221123@163.com (Q.H.)

[2] College of Biological and Agricultural Engineering, Jilin University, Changchun 130022, China; shuyan22@mails.jlu.edu.cn

[3] College of Engineering and Technology, Jilin Agricultural University, Changchun 130118, China

[*] Correspondence: huangdy@jlu.edu.cn

**Abstract:** Rational utilization of soil total nitrogen is one of the keys to achieving sustainable agricultural development. By accurately measuring the content of total nitrogen in the soil, the utilization efficiency of nitrogen in the soil can be improved, and the scientific use of chemical fertilizers can reduce the pressure of agriculture on natural resources and realize the sustainable development of agriculture. In order to measure soil total nitrogen content simply and accurately, combined with the method of artificial olfactory systems, a new method of soil total nitrogen content detection based on convolutional noise reduction autoencoder (CDAE)–whale optimization algorithm (WOA)–deep residual shrinkage network (DSRN) is proposed. In order to obtain more salient features for fusion, the channel mechanism of the DSRN is improved by adding global Max pooling. The model uses a CDAE for the first filtering stage to automatically obtain data that filters simple noise and uses the WOA to automatically optimize hyperparameters. Finally, the optimized hyperparameters were used to train the DRSN for secondary filtering and predict the soil total nitrogen content. Experimental results show that the $R^2$ of CAE-WOA-DSRN test set is 0.968, which is significantly better than the $R^2$ of a traditional algorithm (0.873) and a simple BP network (0.877), and it can more accurately measure soil total nitrogen content.

**Keywords:** soil total nitrogen; artificial olfactory system; prediction method; the depth of the residual shrinkage network; convolution encoder

## 1. Introduction

Soil nitrogen is an essential nutrient for plant growth and development, and it is also an important index to measure soil fertility. Insufficient quantity will not be enough to meet the production demand of regional crops, while excessive quantity will cause environmental problems such as eutrophication of regional water bodies [1]. Therefore, rapid and accurate measurement of soil total N is important for the sustainability of agricultural production.

Currently, chemical detection and spectral detection are the two primary methods for soil nutrient detection. Chemical determination of total nitrogen in soil (including inorganic and organic nitrogen) includes Kjeldahl nitrogen determination, Dumas combustion nitrogen determination, etc. [2–5]. Especially, semi-trace Kjeldahl nitrogen determination is now commonly used [6,7]. Although chemical detection is extensively used in practice and can accurately measure the total nitrogen content in soil, it has some disadvantages such as long time, complicated operation, and pollution.

Spectral detection includes hyperspectral and near-infrared spectroscopy. Hyperspectral remote sensing technology can provide continuous remote sensing imaging of ground objects with very narrow and continuous spectral channels. It has numerous bands, hundreds of spectral channels, and a spectral resolution of up to nanometers. Each spectral

channel is continuous. Adopting this technology is helpful for studying the fine classification and recognition of ground objects by spectral features [8]. However, because of the high cost of its equipment and massive data volume, it is challenging to extract relevant features from it. In addition, the prediction accuracy is relatively low.

With the advantages of fast, accurate, and pollution-free detection, NIR spectroscopy can process a large number of soil samples in a short period of time and realize real-time detection of soil nutrient content [9,10]. However, the spectral method of soil sample assessment is easily affected by soil moisture, iron oxide, soil texture, etc., and its anti-interference ability is poor in practical application [11,12].

In addition, pyrolysis gas chromatography–mass spectrometry (Py-GC/MS) has the advantages of high speed and sensitivity, stable performance, and small sample amount [13]. It also proves that there is a correlation between soil cracking gas and soil nutrients. However, the equipment has the disadvantages of high acquisition cost, professional operation, inability to monitor soil total nitrogen, and time-consuming labor, so it is difficult to achieve rapid measurement of nutrients in a large number of soil samples.

In recent years, electronic nose technology has developed rapidly. For example, ref. [14] used an electronic nose to predict apple freshness and the accuracy reached 0.942. Ref. [15] used an electronic nose to predict the freshness of a refrigerated large yellow croaker and the accuracy reached 0.988. Interestingly, this technology can also be used for soil inspection. In ref. [16], Lavanya et al. used an electronic nose to test the amounts of hyaluronic acid and free fatty acids in soil. In ref. [17], Bieganowski et al. used an electronic nose to realize the evaluation of soil moisture and studied the influence of soil moisture content on the signal of the electronic nose. In refs. [18,19], Longtu and Zhu et al. used a single-sensor array artificial olfactory system to complete the detection of organic matter content. Unfortunately, because only a single-sensor array was used, it could not accurately reflect the chemical reaction of gas; in the literature [20], volatile gases were detected by multiple sensors. However, since the traditional machine learning method combined with a simple backpropagation network was adopted in the regression prediction of gas signal data, it could neither automatically filter the gas signal nor learn the deeper features of the gas signal data. Therefore, the accuracy and response rate of soil total nitrogen content detection still need to be improved.

Deep learning is a more intelligent method than traditional machine learning, which uses the random initialization of neuron parameters of artificial neural networks to automatically extract features, and can learn deeper features in the data as the number of layers of the network deepens. Deep learning is also widely used in regression prediction. However, it also has the problem that the hyperparameter setting is set by the empirical manual, and the hyperparameter is an important factor affecting the final prediction effect of the network, so its efficiency is often very unstable. Furthermore, its interpretability is low due to the automatic nature of the feature extraction.

In summary, traditional methods usually need to collect a large number of soil samples and perform complex experimental analyses, which are time-consuming and costly, or require manual work to extract the features of the data, greatly reducing the efficiency of the overall detection. Therefore, in order to solve the above problems of traditional soil total nitrogen detection methods, such as long detection and analysis time, complex manual operation, high equipment cost, and corrosive reagents. In this paper, the correlation between soil cracking gas and soil total nitrogen is used to make full use of the advantages of low-cost gas sensors. Based on deep learning and a neural network model, a simple, high-precision, and low-cost soil total nitrogen prediction method is proposed so as to realize the rapid, accurate, and low-cost detection of soil total nitrogen content. At the same time, it reduces the need for manual intervention and improves the efficiency and accuracy of the model.

The proposed methodology comprises the following main stages:

1.  The data of soil cracking gas were obtained by thermal cracking and an electronic nose, and the soil cracking gas was analyzed by deep learning methods to predict the soil total nitrogen content.
2.  The convolutional noise reduction encoder is used to reduce the noise and dimensions of the features to reduce the subsequent training time and network parameters.
3.  The whale optimization algorithm adding chaos factor is used to optimize the hyperparameter of the depth residual shrinkage network, and the time to search for the optimal parameter is reduced while automatically searching for the optimal parameter.
4.  An improved channel attention module is introduced into the deep residual shrinkage network, and the feature of global maximum pooling is added to enhance the significance of the feature vector for setting thresholds, so as to achieve a better fitting effect.

## 2. Materials and Methods

### 2.1. Study Area and Soil Sampling

The study area of this paper is an experimental field at the Jilin Academy of Agricultural Sciences in Gongzhuling City, Jilin Province, in the autumn of 2021. A total of 120 soil samples were collected, and the sampling area is shown in Figure 1. This batch of soil samples belongs to black soil in the soil classification system developed by FAO-UN, which is a common cultivated soil type in Jilin Province, with strong swelling, shrinkage, and disturbance characteristics. In order to ensure the randomness and uniformity of the sample, the quincunx sampling method sampling method was used to avoid special locations such as roads, fields, ditches, and manure piles. At each sampling point, 5 topsoil layers (4–20 cm) were taken, and then the 5 soil samples were fully mixed, leaving about 1 kg of soil samples. The samples were sealed and labeled before being brought back to the laboratory and stored in the laboratory with an average room temperature of 24 °C.

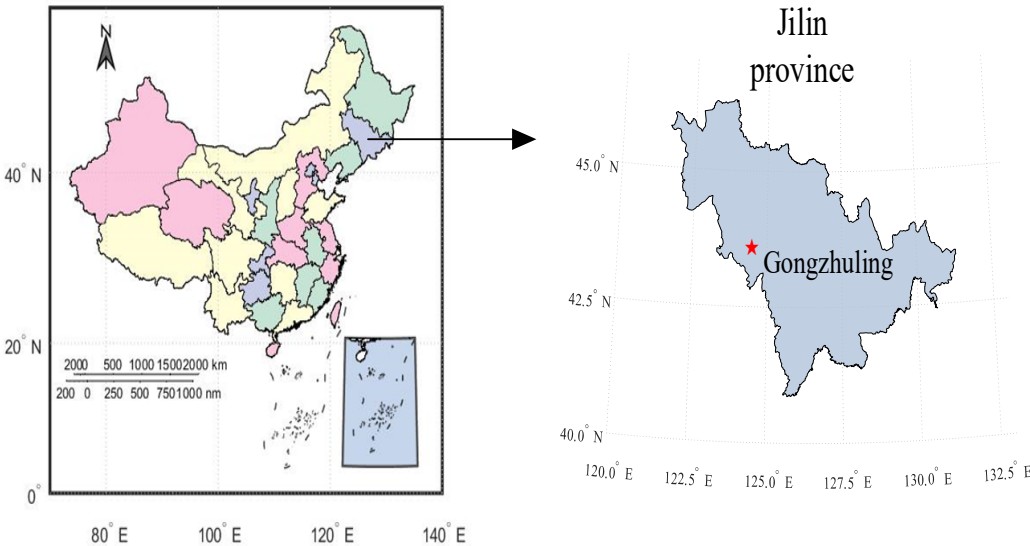

**Figure 1.** Map of the study area: The red star is Gongzhuling.

The collected soil samples were placed in an indoor ventilation place to air dry naturally. The dry soil samples with particles of less than 2 mm were selected through a 70 mesh sieve. Finally, the soil samples were divided into two parts, and the total nitrogen content of the soil samples was determined by the Kjeldahl nitrogen method of the HJ 717-2014 standard [21] and the detection method proposed in this paper [22].

### 2.2. Main Soil Nutrient Detection System

The main soil nutrient detection system devices used in this research include a vacuum flange, tube-cracking furnace, quartz tube, pressure gauge, closed reaction chamber (with

gas sensor array), signal processing circuit, NI data acquisition card, PWM module, vacuum air pump, and computer. The specific structure is shown in Figure 2.

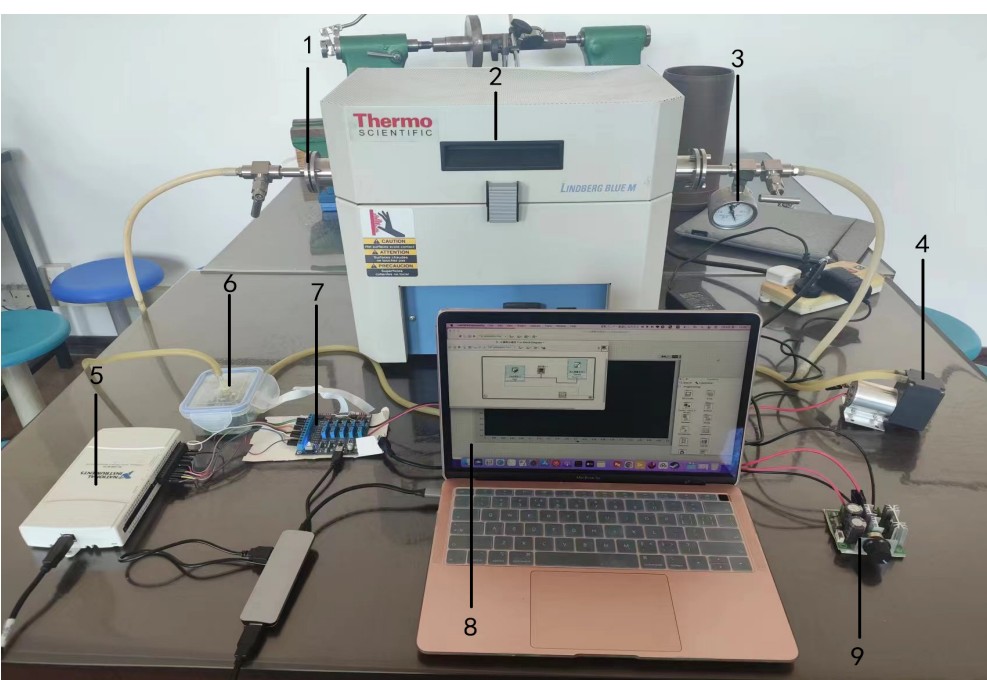

**Figure 2.** Main soil nutrient content detection system device: 1. Vacuum flange; 2. Tube-cracking furnace; 3. Pressure gauge; 4. Vacuum air pump; 5. NI data acquisition card; 6. Closed reaction chamber; 7. Signal processing circuit; 8. Computer; 9. PWM speed regulation module.

In this system, the function of the tube-cracking furnace is to crack the soil sample. A quartz tube containing the soil sample is inserted into the cracking furnace and sealed at both ends by a vacuum flange to form a closed cracking chamber. The manometer monitors the air pressure in the cracking chamber in real time, while the PWM module modulated by the meridian width is used to adjust the flow rate of the vacuum air pump to achieve the gas flow between the vacuum air pump, the cracking chamber, and the closed reaction chamber.

The cracked gas is pushed into the closed reaction chamber equipped with a sensor array through the vacuum air pump, and then through the signal processing circuit, an NI data acquisition card transmits the sensor output signal to the computer. The whole main soil nutrient detection system device can be divided into two parts: the thermal cracking system device and the machine olfactory system device.

The thermal cracking system device is mainly used for cracking soil samples at high temperatures in the central area of the cracking furnace. In order to prevent interference from outside gases, the cracking chamber is kept in a completely closed state as far as possible during the cracking process. The machine olfactory system device is responsible for receiving and processing the cracked soil gas and pushing the cracked gas into the closed reaction chamber through the vacuum air pump to produce a specific response. The analog signal is transformed into a digital signal, and data are acquired by connecting the signal processing circuit to the NI data acquisition card via the Dupont line.

### 2.2.1. Thermal Cracking System Device

The thermal cracking system device includes a tube-cracking furnace, quartz glass tube, vacuum flange, rubber tube, quartz boat, pressure gauge, and other components. In this study, we chose the Lindberg/Blue MTM Mini-MiteTM device manufactured by Thermo Fisher Scientific as the tube-cracking furnace. The cracking furnace is insulated with Moldather material and equipped with a microprocessor self-regulating PID controller,

which can maintain the temperature in the furnace at any value between 100 °C and 1100 °C. The equipment has many advantages, including its long life, small and lightweight nature, single stage, single set point, one-time adjustment to set point, synchronous LED display, adjustable temperature upper limit, open and power-off protection, etc. It is often used in pyrolysis, thermal expansion, calibration, and other experiments. This device ensures rapid rise and fall, high energy efficiency, and relatively low furnace surface temperature.

2.2.2. Machine Olfactory System Device

The machine olfactory system is the detection unit of the whole system, which is composed of three parts, including a gas sensor array, signal processing circuit, and data processing unit. The detection unit structure of the system includes a gas sensor array, reaction chamber, signal processing circuit, NI data acquisition card, computer, and so on.

First of all, the sensor array is the core of the machine olfactory system, which determines the overall performance and detection accuracy of the system from top to bottom.

Therefore, a reasonable choice of sensor types and models is particularly important. Soil total nitrogen includes organic nitrogen and inorganic nitrogen, and soil organic matter is the main source of soil organic nitrogen. During soil thermal pyrolysis, both organic and inorganic nitrogen compounds decompose and release gases. Organic nitrogen compounds will undergo thermal decomposition at high temperatures, when the carbon and nitrogen bonds will break, releasing ammonia, and other organic compounds will release hydrocarbon gases. Similarly, inorganic nitrogen–nitrogen compounds may also decompose during thermal pyrolysis to release gases. For example, ammonium ions can form ammonia [23].

Therefore, depending on the gases released during soil pyrolysis, gas sensors that are likely to produce strong reactions are selected. Considering the practical requirements and applications of volatile gas detection found that the metal oxide semiconductor gas sensor has high sensitivity and high selectivity.

It has the advantages of good performance and fast recovery. Therefore, the MOS semiconductor gas sensor was selected in this study, following the following principles:

(1) High selectivity and sensitivity, strong response to high and low concentrations of target gas;

(2) Wide-spectrum response characteristics, which can realize the concentration detection of a variety of gas molecules;

(3) The response time and recovery time are short, and the test can be repeated in a short time;

(4) In a certain range, non-specific sensor combinations with overlapping response characteristics can be used;

(5) Reusable, high stability, low power consumption, low price.

Based on the above selection principles, the gas sensors selected in this study are mainly the TGS series produced by Figaro, Japan. The model of the gas sensor array selected is shown in Table 1.

**Table 1.** Gas sensor models.

| No. | Model Number | Detect Gas Type | Measuring Range (ppm) |
|---|---|---|---|
| S1 | TGS862 | Ammonia | 30~300 |
| S2 | TGS2602 | Vocs, hydrogen sulfide, etc. | 1~30 |
| S3 | TGS2610 | Butane, LP gas | 500~10,000 |
| S4 | TGS2620 | Ethanol, organic solvent | 50~5000 |
| S5 | TGS821 | Hydrogen | 100~1000 |
| S6 | TGS2603 | Trimethylamine, methyl mercaptan, etc. | 1~10 |
| S7 | TGS2611 | Methane, Natural gas | 500~10,000 |
| S8 | TGS823 | Methane, Ethanol | 50~300 |
| S9 | TGS2600 | Hydrogen, alcohol, etc. | 1~30 |
| S10 | TGS2612 | Methane, propane, isobutane | 3000~9000 |

The output resistance signal of the sensor array is transmitted to the signal processing circuit through the FFC soft bank wire. Next, the signal processing circuit supplies power to the sensor array, while using the principle of resistance voltage division to detect changes in gas concentration and convert the output signal into voltage. The signal processing circuit is connected with the NI data acquisition card by the Dupont line, the data acquisition is carried out, and the analog signal is converted to a digital signal.

Finally, the collected sensor response curve data are sent to the computer through the USB data interface. In the computer, these data are displayed and saved by the LabVIEW application for subsequent data processing. The whole process realizes the perception, acquisition, conversion, and storage of the change in soil cracking gas concentration, which provides convenience for subsequent data processing.

2.2.3. Response Experiment of Soil Nutrient Detection Device

The main soil nutrient detection device was built based on thermal cracking and a machine olfactory system. The complexity of soil composition may result in a wide variety of cracked gases. Therefore, in the initial selection of sensors, it is common to choose as many different types of sensors as possible. However, this could mean that certain gas sensors become less accurate in detecting soil cracking gases, or that certain sensors are less effective in detecting soil cracking gases.

Therefore, this study needs to test the response of soil pyrolysis gas to the sensor of the detection device to verify the rationality of the sensor selection. This testing process will help evaluate the performance of each sensor in practical applications, identify possible response differences or limitations, and provide a basis for further data processing and interpretation. Through this step, the system performance can be optimized and the accuracy and reliability of the test results can be ensured. In order to verify the operational capability of the detector and sensor, we conducted tests on each sensor using the soil sample described in Section 2.1. As shown in the first stage of the response curve of Figure 3, when the soil pyrolysis gas does not enter the reaction chamber, the output voltage of all the sensors remains unchanged in the initial state. This indicates that clean air can be utilized as a cleaning gas because the resistance value of the gas sensor does not alter or respond in a particular way to air.

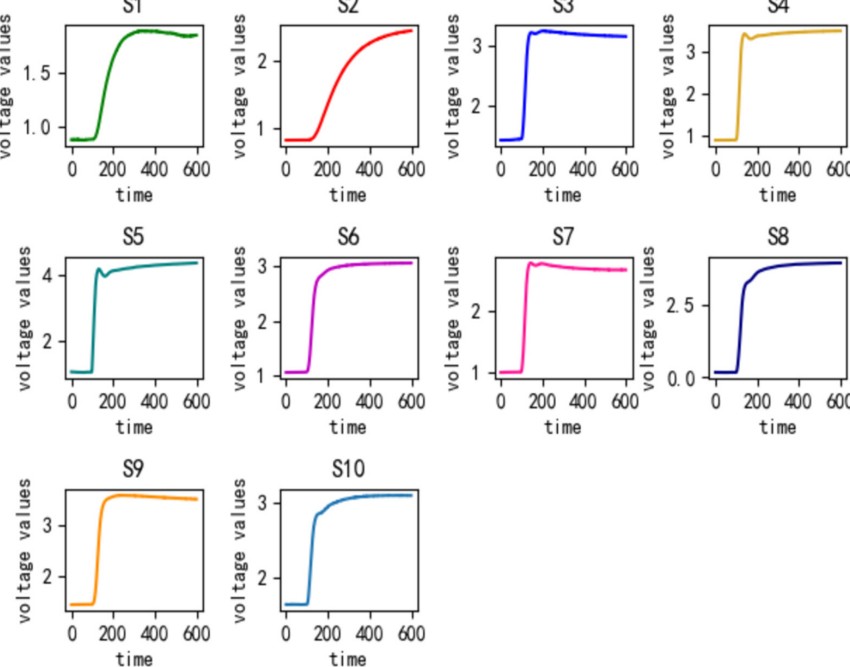

**Figure 3.** Gas signal diagram: S1 to S10 are the ten sensors shown in Table 1.

In the second stage of the response curve, after the introduction of soil cracking gas, all the sensors produced a drastic change in a few seconds. The response curves of the sensors gradually rose. In the following tens of seconds, they all reached a stable state, in line with the normal operation of the sensor mechanism. It can be seen that the selected sensor can produce a specific response to the soil cracking gas, and the built hardware system has a good operating ability. The choice of sensors is quite reasonable.

### 2.3. Data Preprocessing

Firstly, the gas signal data of 600 data points within 60 s after thermal cracking at 500 °C for 2 min were selected, and the values were collected every 0.1 s. The gas signal data with a total of 600 data points are used as the deep learning data set. Figure 3 shows the gas response curves of the 10 sensors. Then, 600 data points such as gas signal response area (Vrav), maximum value of the gas signal (Vmax), average differential coefficient of the gas signal (Vmdc), variance of the gas signal (Vvv), and average value of gas signal were extracted. Six commonly used artificial features of gas signals (Vavg) and maximum gradient of gas signal curves (Vmgv) are used to establish feature spaces to provide data for traditional machine learning methods.

$$Vrav = \sum_{i=1}^{N} X_i \Delta t \tag{1}$$

$$Vmdc = \frac{1}{N-1} \sum_{i-1}^{N} \frac{X_{i+1} - X_i}{\Delta t} \tag{2}$$

$$Vvv = \frac{\sum_{i=1}^{N} (X_i - \overline{X})^2}{N-1} \tag{3}$$

$$Vmgv = \frac{X_{jmax} - X_0}{j} \tag{4}$$

In the formula calculation, the meanings of Vrav, Vmdc, Vvv, and Vmgv are as follows.

$X_i$ represents the i-th data point collected by the sensor; $X_i \Delta t$ represents a 0.1 s time interval; N represents the number of total points; $\overline{X}$ represents the average of the data; $X_0$ represents the initial value; j represents the time corresponding to the maximum value.

In order to eliminate the influence of dimension and order of magnitude on the prediction results and accelerate the speed of gradient descent, this study adopts the Standscaler method, namely normal distribution standardized z-score, to process the data.

$$z = (x - \mu)/s \tag{5}$$

where z represents a single standardized data point; x represents a single raw data point; $\mu$ represents the total raw data average; s represents the standard deviation of the overall raw data.

### 2.4. Division of Training Set and Test Set

A total of 120 data sets are used in this paper, each of which contains 61 values. The first 60 values are the above 10 sensors, and the 6 characteristic values are extracted from each sensor. The 61st value is the true value obtained from the chemistry test, as shown in Table 2, and the 6 characteristic values and true values of sensor 1. Among them, the chemical test of soil nitrogen content, the label of this data set, comes from "Product and Processed Product Quality Supervision and Inspection Testing Center of Ministry of Agriculture and Rural Affairs (Changchun)".

**Table 2.** Data set format for total nitrogen content of soil samples: TN is total nitrogen. Remaining eigenvalues are described within Section 2.3.

| | Vrav/(g·s·kg$^{-1}$) | Vmax/(g·kg$^{-1}$) | Vmdc/(g·(kg·s)$^{-1}$) | Vf/(g·kg$^{-1}$) | Vavg/(g·kg$^{-1}$) | Vmgv/(g·(kg·s)$^{-1}$) | TN/(g·kg$^{-1}$) |
|---|---|---|---|---|---|---|---|
| Sensor 1 | 2.4 | 4.2 | 3.4 | 3.6 | 1.9 | 2.2 | 0.91 |

In this study, the ratio of the training set to the test set was set at 8:2. More specifically, the training set consisted of 96 samples and the test set consisted of 24 samples.

### 2.5. Model Construction

2.5.1. Whale Algorithm and Chaos Mapping

The whale optimization algorithm (WOA), published in 2016 by Mirjalili and Lewis [24], is a heuristic optimization method that draws on the behavior patterns of whale clusters in nature. This algorithm mainly mimics the hunting mode and strategy of humpback whales, covering three core hunting behaviors [25], namely prey rounding, bubble net hunting, and prey searching.

In this paper, the whale optimization algorithm is used to optimize each hyperparameter of the model.

The traversal of the Tent map has uniformity and randomness, which can make the algorithm converge faster. This paper uses Tent mapping to generate chaotic sequences and initialize the population, so that the initial solution is distributed as evenly as possible in the solution space. The Tent sequence was adopted in this paper, where the initial value of $\omega_0$ is a random number from 0 to 1, and the subsequent sequence number is obtained according to Formula (6). The high quality of the initial population is of great help to the performance of the algorithm, such as convergence speed and solution accuracy [26].

$$\omega_{t+1} = \begin{cases} 4 \times \omega_t & \omega < 0.5 \\ 4 \times (1 - \omega_t) & \omega \geq 0.5 \end{cases} \tag{6}$$

2.5.2. CDAE Model

Convolution Auto Encoder (CAE) is a special case of traditional autoencoders, and it uses convolutional layers and pooling layers instead of the original fully connected layer. The convolution encoder structure consists of an encoder and a decoder, the encoder consisting of multiple convolution and pooling layers, and the decoder consisting of multiple deconvolution and sampling layers. The convolution layer can extract the local features of the input data, the pooling layer can reduce the dimensional feature map, the deconvolution layer can restore low-dimensional features to high-dimensional features, the bottleneck layer can reduce the spatial dimension of the feature map while keeping the number of input data channels unchanged, thereby reducing the computational cost of the model and increasing the nonlinear characteristics of the network, and the sampling layer can restore low-resolution features to high-resolution features.

The convolutional noise reduction encoder replaces the input data with the data with noise, and uses the error between the reconstructed data and the data without noise as the loss function. In this way, decoding parameters for removing noise can be obtained to achieve the purpose of noise reduction. A convolutional noise reduction autoencoder (CDAE) is formed. In this paper, gas signal data with white Gaussian noise are used as noise data, as shown in Figure 4. After inputting the gas signal, it is converted into 64 channels through a convolution layer, and then the signal length is compressed to half of the original. After that, the signal features of $4 \times 70$ are obtained through the last convolution pooling. After flattening, the features compressed to $1 \times 1 \times 70$ by a linear layer are used as the dimensionality reduction features [27].

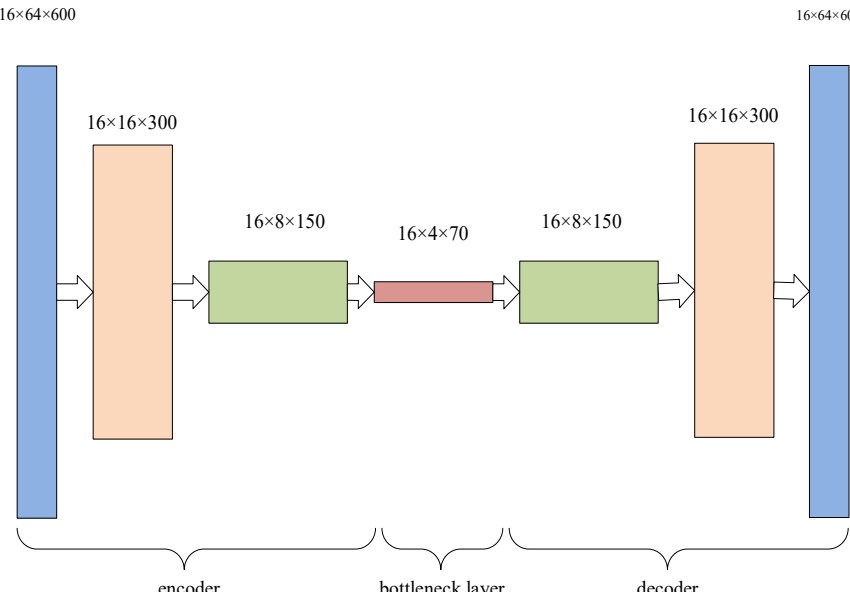

**Figure 4.** Convolutional autoencoder structure diagram.

### 2.5.3. Improved Deep Residual Shrinkage Network

The deep residual shrinkage network (DSRN) is an improvement of one-dimensional convolutional neural networks. Compared with the ordinary one-dimensional convolutional neural network, it not only solves the problem of gradient explosion and gradient disappearance through identity mapping but also eliminates noise-related features in data through the principle of soft thresholding [28].

In this paper, the soft threshold is transformed into an inserted structural unit and combined with the improved channel attention mechanism to automatically set the threshold. The original structure is used for fault identification (classification) of vibration signals, so it is more valuable to use global average pooling to obtain average features for the final classification. However, the goal of this study is to predict features according to the content of gas signals. The average features can be used for noise reduction in the stable stage, but the average features will cause certain feature losses in the rising stage. Therefore, it is proposed to add significant features to make up for part of the feature loss.

In order to extract more relevant features, a parallel global maximum pooling module is added to the original component, which only employs the global average pooling channel attention technique to minimize the dimensionality of the features recovered from the convolutional layer. In the actual module, GAP is used to obtain the absolute value of the feature space x, and MAP is used to obtain the maximum value of the feature space x, to obtain two one-dimensional vectors representing the overall feature of the signal as a threshold.

Subsequently, the two one-dimensional vector features are passed into two fully connected layers, namely the multilayer perceptron, respectively, to obtain the scaling parameters. After the two fully connected layers, the two scaling parameters are first added and fused, and a sigmoid function is used, at which time the scaling parameters are converted to a value in (0, 1). This scaling parameter is then multiplied by the mean value of the feature graph |x| to obtain the threshold for filtering. After the threshold value is obtained, the corresponding soft threshold processing can be conducted through the number multiplication operation of the matrix. Finally, the final output of the module is obtained through the residual connection and input addition. The specific module structure diagram is shown in Figure 5a. is the improved module structure, and Figure 5b. is the original module structure.

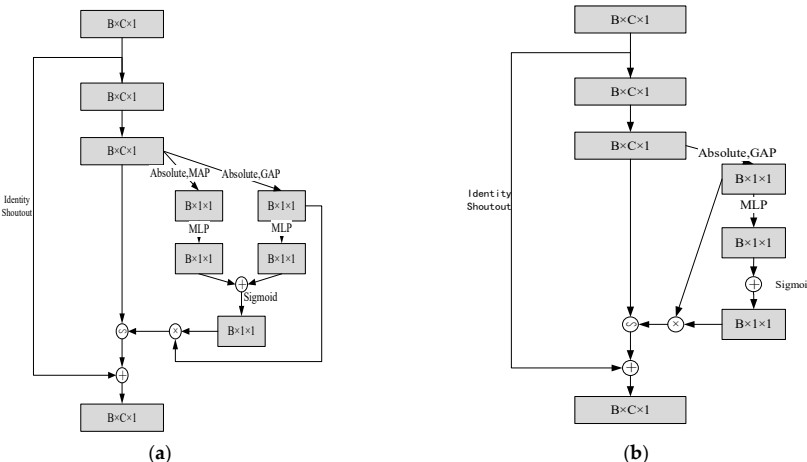

(a)  (b)

**Figure 5.** (**a**) Improved module structure; (**b**) original module structure, where + represents matrix addition fusion, × represents number multiplication operation, and ⌣ represents number multiplication operation of the matrix.

In order to enable the entire network to adjust to scenarios in which a sample contains multiple complex noises, an automatic channel threshold module with deep residual shrinkage and an improved channel attention mechanism are employed in this study [29]. This means that a threshold is set for each channel, indicating that every sample receives a separate threshold.

The overall network architecture of the deep residual shrinking network used in this study was modified by the network architecture of ResNet. After inputting the signal data with a size of Batch_size × Channel × 1 and passing them through a one-dimensional convolution layer, the channel of the input signal is converted into 16 channels, and the parameters are reduced through a global average pooling. The basic module of ResNet is replaced by the above module as the basic module of the deep residual shrinkage network. Then, the module is stacked twice as a convolution layer, and a pooling layer is added after every two convolution layers to reduce the parameters. Each layer will compress Batch_size to half while doubling the number of channels, so after three convolution layers, the results are input to the Flatten layer for the flattening operation. And then the final predicted result can be achieved by inputting them into the linear layer, as shown in Figure 6.

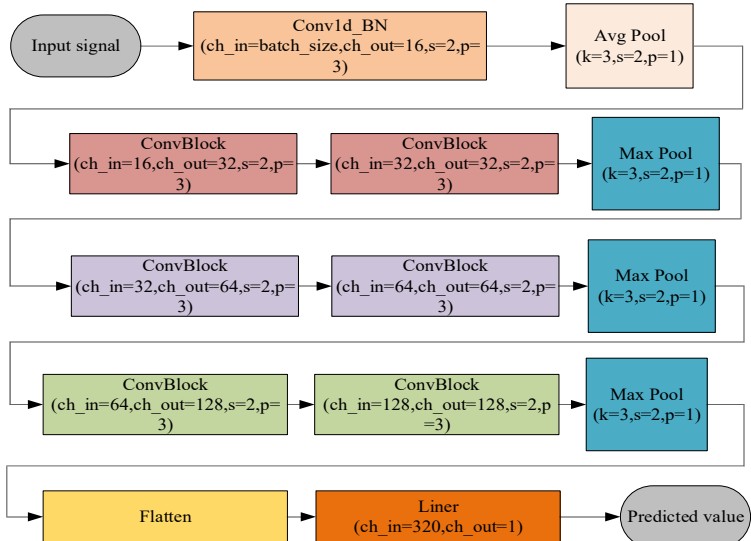

**Figure 6.** Depth residual network structure diagram.

### 2.5.4. CDAE-WOA-DRSN Model

In order to predict the accuracy of total nitrogen content, this paper uses the improvement and combination of CDAE, DSRN, and WOA algorithms to build a prediction model between artificial olfactory feature space and soil total nitrogen content, aiming to find the best correlation model. The overall processing flow is shown in Figure 7.

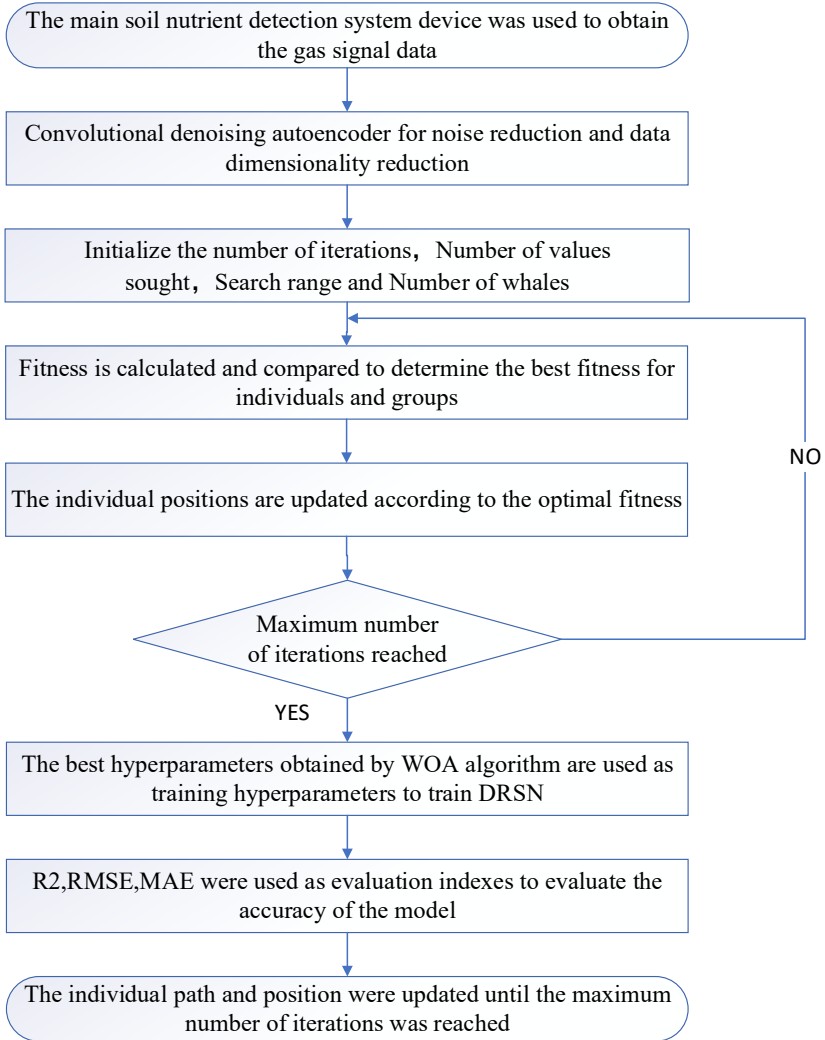

**Figure 7.** Flow chart of the algorithm.

Step 1: Data acquisition part. Use the soil major nutrient detection system device described in Section 2 to prepare the soil cracking gas and obtain the raw gas signal data.

Step 2: Data processing section. The convolutional noise reduction autoencoder is used to process the original data into $1 \times 1 \times 70$ feature data and divide them into corresponding training data and test data.

Step 3: Train the model. Use the training data to train the model and use the improved WOA to optimize the hyperparameters that affect the model's training effectiveness.

Step 4: Prediction. Train the model using the optimized hyperparameters and predict the total nitrogen content of the soil based on the test data.

Step 5: Evaluate the comparison. The prediction results and the actual results are used to calculate the relevant evaluation indicators, and the most suitable model is selected for the prediction of soil total nitrogen content.

*2.6. Model Evaluation Index*

$$\text{MAE} = \frac{1}{n}\sum_{i=1}^{n}\left|\hat{y}_i - y_i\right| \tag{7}$$

$$\text{RMSE} = \sqrt{\frac{1}{n}\sum_{i=1}^{n}\left(\hat{y}_i - y_i\right)^2} \tag{8}$$

$$R^2 = 1 - \frac{\sum_i\left(\hat{y}_i - y_i\right)^2}{\sum_i\left(\bar{y}_i - y_i\right)^2} \tag{9}$$

In order to verify the accuracy and stability of the network model, three evaluation indexes, namely mean absolute error (MAE), root mean square deviation (RMSE), and coefficient of determination ($R^2$), were selected as the basis for evaluating the performance of the model.

MAE is the mean absolute error of the network model; RMSE is used to represent the error between the predicted value and the true value of the network model, that is, the closer MAE and RMSE are to 0, the higher the accuracy of the network prediction. And $R^2$ represents the interpretation ability of the model to the dependent variable, that is, the closer $R^2$ is to 1, it indicates that the fitting function of the network is closer to the real situation, and the prediction result is closer to the real value.

## 3. Results

*3.1. Total Nitrogen Content Statistics of Soil Samples*

The descriptive characteristics of total nitrogen in 120 soil samples measured are shown in Table 3. The content of total nitrogen in soil ranges from 0.20 to 4.10 g·kg$^{-1}$, with an average value of 1.59 g·kg$^{-1}$, standard deviation of 0.73 g·kg$^{-1}$, coefficient of variation of 45.77%, skewness of 1.17 g·kg$^{-1}$, and kurtosis of 2.21 g·kg$^{-1}$. The variation trend of soil total nitrogen content collected in the study area and the coefficient of variation of the samples were large, indicating that there were large differences in total nitrogen content among the soil samples in the study area, which was conducive to the prediction of the subsequent model.

**Table 3.** Statistics of total nitrogen content in soil samples.

|  | Max (g·kg$^{-1}$) | Min (g·kg$^{-1}$) | Avg (g·kg$^{-1}$) | Std (g·kg$^{-1}$) | CV (%) | Sk (g·kg$^{-1}$) | Ku (g·kg$^{-1}$) |
|---|---|---|---|---|---|---|---|
| TN | 0.20 | 4.10 | 1.59 | 0.73 | 45.77 | 1.17 | 2.21 |

*3.2. Experimental Results*

The generalization ability of the DSRN is mainly related to data set size, data noise interference degree, model structure, module layers, iteration times (EPOCH), learning rate (LR), batch size (Batch_size), and other hyperparameters.

Since there are only 120 samples in this experiment, the network contains the Batch Normalization (BN) layer of dropout, and the effect of adding the dropout layer is not good, in order to reduce the degree of overfitting, we need to adopt a shallow network structure. At the same time, in order to improve the feature extraction ability of each layer, a large convolution kernel is adopted for feature extraction. After many experiments, the results are shown in Table 4. It can be found that when the convolution kernel size is 7, 31, and 7, respectively, Test_$R^2$ is the largest, and Test_RMSE and Test_MAE are the smallest, that is, the prediction effect is the best. Therefore, the 1D-DSRN model in this paper uses each basic convolutional module containing three convolutional layers, using convolution kernel sizes of 7, 31, and 7, respectively, to extract features, and with the deepening of the

number of layers, according to the number of channels, 16, 32, and 64 to extract features in turn.

**Table 4.** Influence of convolution kernel size on the performance of the 1D-DSRN model.

| Convolution Kernel 1 | Convolution Kernel 2 | Convolution Kernel 3 | Test_$R^2$ | Test_RMSE | Test_MAE |
|---|---|---|---|---|---|
| 1 | 3 | 1 | 0.935 | 0.253 | 0.196 |
| 1 | 7 | 1 | 0.893 | 0.326 | 0.288 |
| 3 | 7 | 3 | 0.948 | 0.226 | 0.184 |
| 1 | 31 | 1 | 0.926 | 0.270 | 0.230 |
| 3 | 31 | 3 | 0.902 | 0.311 | 0.254 |
| 7 | 31 | 7 | 0.951 | 0.221 | 0.183 |

In order to enable the network to extract correct features from complex noise, this paper constructs a noise reduction encoder through a CDAE as the first filter and reduces the dimension. Secondly, the channel attention mechanism used to set the threshold value in the DRSN is improved, and the original global maximum pooling is added to the global average pooling, so that the network not only extracts the most obvious features in the sample but also extracts the features of the whole sample, that is, more general features. In this way, more noise can be removed from the final threshold setting, which can improve the accuracy. And then the two pooled feature vectors are fused as the weight of the threshold setting to set the threshold value.

After several experiments, the results are shown in Table 5. It is found that the fused feature vectors after global maximum pooling and global average pooling are multiplied by the mean value of the feature graph $|x|$ to obtain a threshold with a higher accuracy of 0.951, while the predicted accuracy of the threshold vector by multiplying the fused feature vector with the maximum value of the feature graph $|x|$ is 0.937, which is lower than the threshold obtained by multiplying the average value of the feature graph $|x|$. Therefore, in the end, the feature vectors fused after global maximum pooling and global average pooling are selected as weights to multiply with the average value of the feature graph $|x|$, and the threshold obtained by weight balancing is taken as the final threshold for processing.

**Table 5.** Influence of thresholds set by different pooling methods on the performance of the 1D-DSRN model.

|  | $R^2$ | RMSE | MAE |
|---|---|---|---|
| GAP × AVG($|x|$) | 0.945 | 0.233 | 0.181 |
| MAP × AVG($|x|$) | 0.927 | 0.268 | 0.229 |
| (GAP + MAP) × AVG($|x|$) | 0.951 | 0.221 | 0.183 |
| GAP × MAX($|x|$) | 0.917 | 0.286 | 0.238 |
| MAP × MAX($|x|$) | 0.941 | 0.242 | 0.205 |
| (GAP + MAP) × MAX($|x|$) | 0.937 | 0.251 | 0.202 |

In order to improve the accuracy of fitting and due to the complexity of manual adjustment parameters, EPOCH, LR, and BATCH_SIZE are optimized by the whale optimization algorithm. In order to verify the effectiveness of the whale optimization algorithm, the empirical manual adjustment parameters EPOCH, LR, and BATCH_SIZE are compared with the EPOCH, LR, and BATCH_SIZE optimized by the whale optimization algorithm to set the network parameters. The determination coefficient is taken as the evaluation index. Additionally, since EPOCH and BATCH_SIZE must be integers during the network operation, BATCH_SIZE must be greater than 1 at the BN layer. And the whale optimization algorithm in the process of calculating the value of the generated value is a floating point number, so the two hyperparameters were rounded up to ensure that the value is an integer greater than 1. The results are shown in Table 6. The first six behaviors are

optimized according to the empirical manual adjustment parameters, and the last line is optimized by the algorithm parameters. It can be found that the effect of the parameters optimized by the algorithm is indeed better, and the parameter values are also the values rarely obtained by the general empirical manual parameters.

**Table 6.** Effects of different hyperparameters on the model ability of 1D-DSER.

| EPOCH | LR | BATCH_SIZE | $R^2$ | RMSE | MAE |
|---|---|---|---|---|---|
| 1000 | $1 \times 10^{-3}$ | 64 | 0.951 | 0.221 | 0.183 |
| 800 | $1 \times 10^{-3}$ | 64 | 0.951 | 0.221 | 0.183 |
| 800 | $1 \times 10^{-2}$ | 64 | 0.896 | 0.321 | 0.263 |
| 800 | $1 \times 10^{-3}$ | 16 | 0.950 | 0.221 | 0.184 |
| 800 | $1 \times 10^{-3}$ | 32 | 0.937 | 0.250 | 0.21 |
| 800 | $1 \times 10^{-3}$ | 128 | 0.854 | 0.381 | 0.315 |
| 647 | $1 \times 10^{-3}$ | 84 | 0.957 | 0.207 | 0.164 |

### 3.2.1. Model Extraction Features

After building the model and selecting the optimal hyperparameters, the features extracted by the two methods were visualized separately in order to compare the differences between features extracted by traditional methods and deep learning. As shown in Figure 8, the values of S3, S6, and S7 are very different, and the values of features extracted by deep learning in S3 and S6 are dozens of times higher than those extracted by traditional methods. However, the value of S7 is tens of times higher than the traditional method, which may be an important factor leading to the final result.

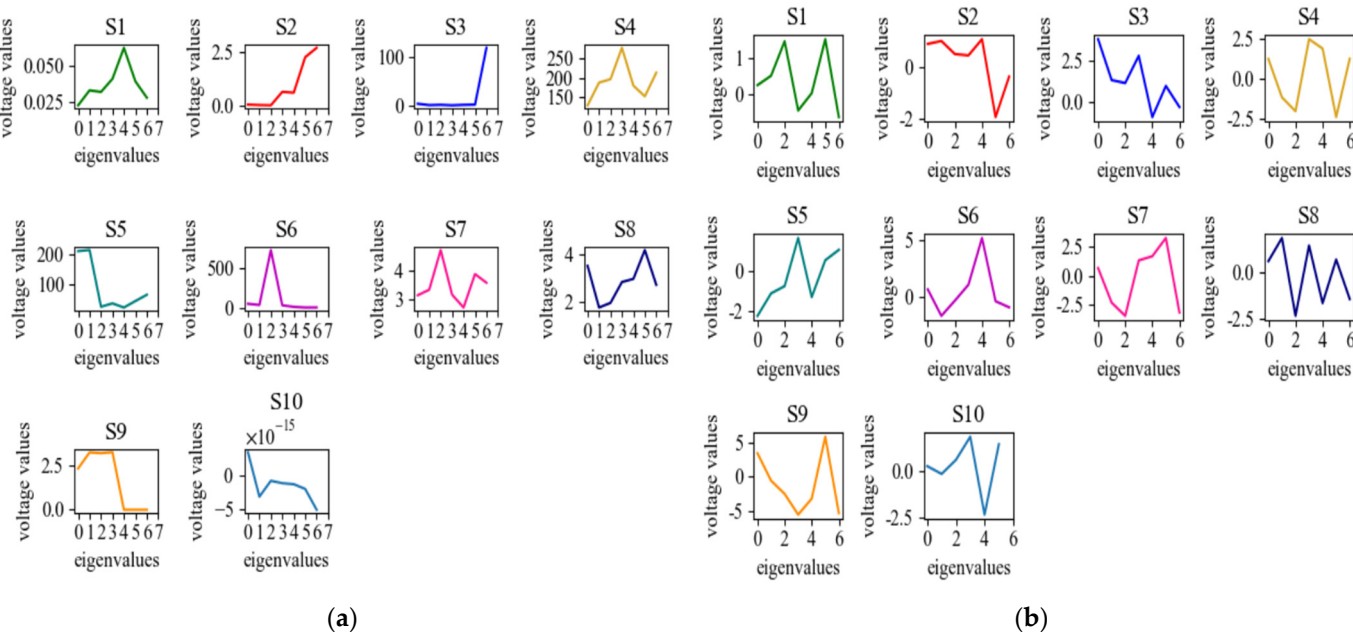

**(a)**　　　　　　　　　　　　　　　　　　　　　**(b)**

**Figure 8.** Feature signal diagram after pretreatment: (**a**) features extracted by convolutional noise reduction autoencoder; (**b**) features extracted according to Formulas (1)–(4).

### 3.2.2. Comparison of Models

In order to select the most effective prediction model and improve the versatility of the soil total nitrogen olfactory detection model, PLSR, SVR, and RF, which are typical of traditional machine learning, were selected and verified, respectively. In order to ensure the synchronization of the training set and test set with the training set and test set randomly generated by the deep learning method, the TRAIN_TEST_SPLIT method in sklearn was used to divide the data set, and the random number seed was set to 99. In addition, BPNN,

which represents a neural network, CNN, DSRN, and other classical one-dimensional regression prediction models in deep learning algorithms are used for comparison. The results are shown in Figure 9.

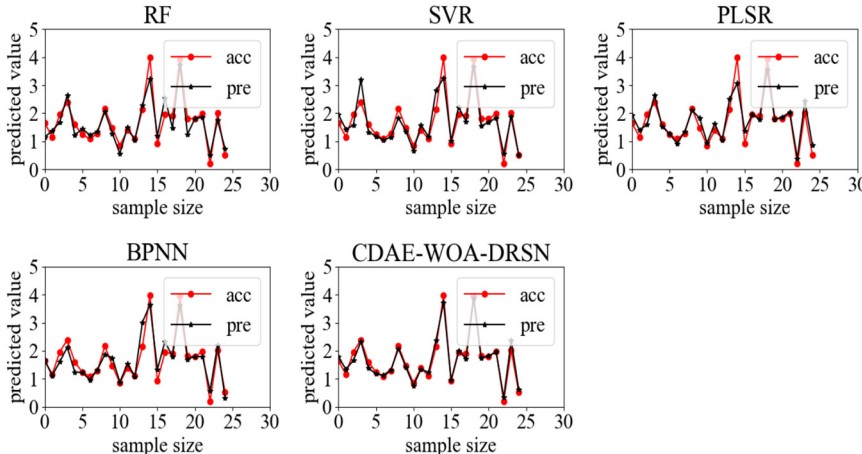

**Figure 9.** Prediction results of each model.

The main parameters of the RF algorithm are the decision tree and the number of leaves. If the number of decision trees is too large, the calculation time will be affected. However, if the number is too small, the regression prediction effect will be reduced. Leaves are the end nodes of the decision tree, and a too-small number of leaves will make the model more susceptible to noise in the data [30]. In this paper, it is found through several tests that the best effect is achieved when the number of decision trees is 3, the minimum number of leaves is 2, and the maximum number of leaves is not limited. $R^2$ is 0.858, and it can be seen that its evaluation index is relatively low, probably because RF is more suitable for classification tasks, and is not suitable for the specific numerical prediction of this task.

The main influencing parameters of the SVR algorithm are the penalty factor C and the kernel function parameter g. The larger C is, the more attention is paid to the total error in the whole optimization process, and the higher the requirement is for error reduction. When C tends to infinity, no sample of error is allowed to exist. When C approaches 0, only a larger interval is required. No meaningful solution can be obtained and the algorithm will not converge. The g value must be greater than 0, and with an increase in the g value, the higher the complexity of the model, the worse the generalization ability, and the higher the overfitting degree [31]. In this paper, it is found through many experiments that the best effect is $R^2 = 0.871$ when kernel = 'poly', C = 1, and g = 0.48. Compared with the RF algorithm, its evaluation index is improved, although its model rating is unchanged. Other indicators are better than the RF algorithm, indicating that the SVR model has better performance and tends to be more stable in the specific numerical prediction of this task.

The main Influencing parameter of the PLSR algorithm is the number of its principal components. If the number of principal components is too large, the prediction effect will be better, but it will lead to overfitting of the model. If the number of principal components is too small, the complexity of the model will be reduced, but the prediction effect will also be reduced [32]. In this paper, it is found through many experiments that when the number of principal components is 5, the best effect is achieved, and $R^2$ is 0.873. Compared with the SVR algorithm, its prediction effect and fitting degree are improved, but the improvement is not large. The PLSR algorithm and the SVR algorithm may be more suitable for the numerical prediction required by this task than the RF algorithm.

The summary of the model comparison is shown in Table 7. To sum up, among the three traditional machine learning algorithms (PLSR, SVR, and RF), PLSR has the best predictive performance ($R^2$ is the largest, RMSE and MAE are the smallest), SVR has the second-best predictive effect, and RF has the lowest predictive performance.

**Table 7.** Prediction effects of different models on soil total nitrogen.

| Prediction Models | $R^2$ | RMSE | MAE |
|---|---|---|---|
| RF | 0.858 | 0.320 | 0.267 |
| SVR | 0.871 | 0.306 | 0.239 |
| PLSR | 0.873 | 0.303 | 0.228 |
| BPNN | 0.877 | 0.301 | 0.227 |
| CNN | 0.907 | 0.295 | 0.219 |
| DSRN | 0.929 | 0.289 | 0.232 |
| CDAE-DSRN | 0.945 | 0.236 | 0.190 |
| WOA-DSRN | 0.957 | 0.207 | 0.164 |
| CDAE-WOA-DSRN | 0.968 | 0.176 | 0.176 |

In neural networks and deep learning models, the main parameters that affect the model are various hyperparameters of the model and the number of hidden layers of the network, such as EPOCH, LR, BATCH_SIZE, etc.

When EPOCH BPNN is 50, LR is 1E-2, and BATCH_SIZE is 70, the optimal effect can be achieved. The prediction result with an $R^2$ of 0.877 is shown in Figure 10. It can be seen from the figure that although its evaluation index is higher than that of traditional machine learning algorithms, it is not much higher, which may be inferred because the BPNN is not a deep learning algorithm. The hidden layers in its network structure can only be used with shallow layers to reduce overfitting caused by overcomplexity of the model, so not enough features are automatically learned to fully fit a function curve that predicts total nitrogen content [33].

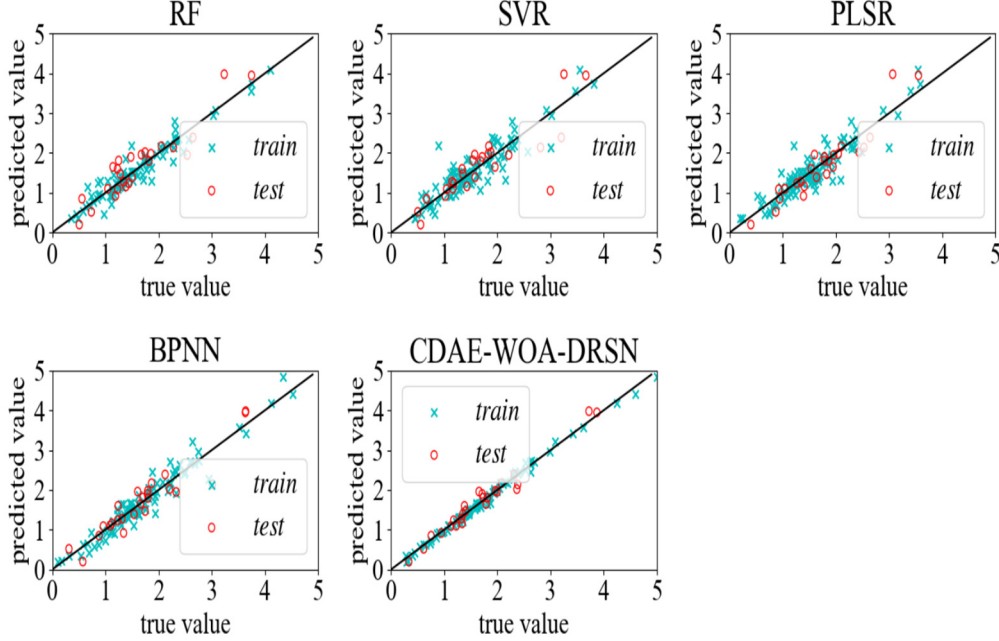

**Figure 10.** Figure of the prediction results of each model.

By comparing the effect of a DSRN with the effect of a CNN in deep learning algorithms, it can be found that the $R^2$ of the DSRN is 0.929, which is better than the $R^2$ of the CNN (0.907) and is also in line with the experiment of Minghang et al. [29]. In their work, it was found that the effect of the CNN was worse than that of DSRN when there was noise interference. This may be due to the existence of certain noise in the data. So the denoising processing was further added to carry out the experiment.

By comparing the effect of a CDAE and DSRN, it was found that the noise reduction effect of Gaussian white noise added to the CDAE was better ($R^2$ = 0.945), which proved that the noise reduction treatment had indeed improved the prediction effect.

Finally, the WOA was combined with CDAE-DSRN to optimize its parameters, and the best effect was achieved when Max_iter = 5, dim = 3, SearchAgents = 5, and $R^2$ = 0.968.

In summary, among neural networks and deep learning models, the prediction effect of the 1D-CDAE-WOA-DSRN proposed in this paper is the best, followed by the DSRN optimized by the whale optimization algorithm, namely the 1D-WOA-DSRN, followed by the 1D-CDAE-DSRN with a convolutional noise reduction autoencoder added. While the effect of the DSRN without the WOA is the same as that of the CNN, the BPNN has the worst effect but its $R^2$ is greater than 0.87, indicating that the five models have good prediction ability and the effect is better than that of traditional machine learning methods.

In order to obtain the best model and verify the stability of the model, a more intuitive method is adopted to compare the fitting results of various models. The results are shown in Figure 10. It can be seen from the figure that the fitting degree of the CDAE-WOA-DSRN proposed in this paper is the best, and the training set and test set are both near the fitting line and intersect with the fitting line.

Among the three traditional learning algorithms, the RF algorithm has data points in the test set that are farther away from the fitting line than those in the training set, showing a fitting trend. Although the PLSR and SVR algorithms did not select the trend of overfitting, the data points in their test set had many data points far away from the fitting line compared with the data points in the CDAE-WOA-DSRN, and their fitting degree was far lower than that of the algorithms in the CDAE-WOA-DSRN.

However, although the BPNN has a better effect than the other three traditional algorithms, the three traditional algorithms are not accurate in fitting the data point of acc = 3.2 in the test set. On the contrary, both the BPNN and CDAE-WOA-DSRN are accurate in fitting the data point. It is speculated that the artificially extracted features for this data point cannot describe these data well. In addition, the situation in which the training data within 3 g·kg$^{-1}$ in the training set of the three traditional algorithms all have obvious outliers is further advanced, but there is still a big gap compared with the effect of the basic data points of the CDAE-WOA-DSRN all having intersection points with the fitting line.

## 4. Discussion

Nitrogen content in soil directly affects crop growth and yield. Through the rapid and accurate assessment of soil total nitrogen content, farmers can optimize the fertilizer application rate, improve fertilizer utilization rate, reduce environmental pollution, provide a scientific basis for agricultural production, and achieve sustainable development. Therefore, accurate knowledge of soil nitrogen content is crucial to achieve efficient agricultural production.

The traditional methods for the determination of soil total nitrogen are time-consuming, and the reagents used are corrosive. The determination method of soil total nitrogen by near-infrared spectroscopy is affected by soil texture, soil moisture, and iron oxide. The Py-GC/MS method has the disadvantages that it has a high equipment purchase cost, cannot be dedicated to the determination of soil total nitrogen, and is time-consuming and labor-intensive, so it is difficult to realize the rapid measurement of total nitrogen content in a large number of soil samples.

During soil pyrolysis, organic compounds and inorganic nitrogen compounds decompose to release gases.

Therefore, by monitoring the gases produced by soil pyrolysis, we can indirectly understand the content and distribution of organic and inorganic N in the soil. Specific types of gas sensors were selected to monitor the gases produced by soil pyrolysis based on the aforementioned gases, due to the high sensitivity and selectivity of these sensors for the detection of specific gases. They are able to accurately identify nitrogen compounds in the soil gas, thus helping us to understand the nitrogen status of the soil and predict the total nitrogen content of the soil.

Therefore, a variety of traditional statistical methods and deep learning techniques were used to analyze the gas produced by soil pyrolysis and predict the soil total nitrogen content. These methods include traditional algorithms such as PLSR, SVR, RF, BPNN, and the deep learning model CDAE-WOA-DSRN. Using MAE, RMSE, and $R^2$ as indicators, the $R^2$ obtained is more than 0.85. It shows that there is an inherent law between soil cracking gas and soil total nitrogen content.

Finally, the best model CDAE-WOA-DSRN was selected based on the performance and accuracy of different algorithms to ensure an accurate assessment of soil N status.

## 5. Conclusions

In summary, this study proposed a method for soil total nitrogen content detection based on thermal cracking and an electronic nose. The thermal cracking technology was used to achieve rapid cracking of soil samples, and the electronic nose was used to complete the data collection of cracked gas responses. Finally, the deep learning neural network model was used to accurately predict the soil total nitrogen content.

A new method, CDAE-WOA-DSRN, combined with a convolutional denoising autoencoder (CDAE), the whale optimization algorithm (WOA), and the deep residual shrinkage network (DSRN), was proposed to detect soil total nitrogen content. Combined with the artificial olfactory technique, the CDAE was used for preliminary data filtering, the WOA was used for the automatic optimization of hyperparameters, and the DSRN was used for secondary filtering and prediction of soil total nitrogen content. The experimental results show that the $R^2$ value of the CDAE-WOA-DSRN on the test set reaches 0.968, which is significantly better than the traditional algorithm and simple BP network, proving that the CDAE-WOA-DSRN can measure the soil total nitrogen content more accurately.

The results of this study have important theoretical and practical significance.

(1) This method can achieve the simple and accurate measurement of soil total nitrogen content, thereby improving soil nitrogen use efficiency, promoting scientific fertilization, reducing the pressure of agriculture on natural resources, and promoting the sustainable development of agriculture.

(2) The combination of a CDAE and the WOA can realize automatic hyperparameter optimization, reduce the need for manual intervention, and improve the efficiency and accuracy of the model.

(3) The introduction of DSRN further optimized the prediction ability of the model, making it perform well in the prediction of soil total nitrogen content. Therefore, the CDAE-WOA-DSRN method proposed in this study not only provides an innovative solution for the evaluation of soil nitrogen but also provides an idea for the rapid measurement and prediction of other soil nutrient components.

**Author Contributions:** Conceptualization, H.L. and D.H.; methodology, J.W. and H.L.; software, J.W.; validation, H.L., D.H. and S.L.; formal analysis, D.H.; data curation, D.H.; writing—original draft preparation, J.W.; writing—review and editing, H.L., Q.H. and S.L.; supervision, H.L.; visualization, J.W., S.L. and Q.H.; funding acquisition, H.L. and D.H. All authors have read and agreed to the published version of the manuscript.

**Funding:** This research was financially supported by the National Key Research and Development Program (grant number 2023YFD1500404) and the Innovation Platform and Talent Special "Agricultural Image Recognition and Processing Team" of the Jilin Science and Technology Department (grant number 20220508133RC).

**Institutional Review Board Statement:** Not applicable.

**Informed Consent Statement:** Not applicable.

**Data Availability Statement:** The data presented in this study are available from the corresponding author upon reasonable request.

**Conflicts of Interest:** The authors declare that they have no known competing financial interests or personal relationships that could have appeared to influence the work reported in this paper.

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
