# Peer review of "Study on Soil Total Nitrogen Content Prediction Method Based on Synthetic Neural Network Model"

_sustainability, doi:10.3390/su16083195_

Round 1
Reviewer 1 Report
Comments and Suggestions for Authors
Recommendations to authors
• In line 2: I believe the title could be improved. My suggestion is to replace the abbreviation CDAE-WOA-DSRN with a term that would make the title clearer.
• In line 15: You must explain what “CDAE-WOA-DSRN” stands for on its first appearance. Therefore, please check first if it is correct and then add in the text the following "Convolutional noise reduction auto-encoder (CDAE) - Whale Optimization Algorithm (WOA) – Deep Residual Shrinkage Network (DSRN)". The term "DSRN" is not explained anywhere in the text.
• In lines 19-20: R2 must be "R2". Please check and make corrections throughout the text.
• In line 89: Maybe instead of "The main contributions of this paper are summarized as follows:" you could use " The proposed methodology comprises the following main stages:".
• In line 100: While you describe the steps of the proposed methodology, you do not mention the specific problem you are attempting to address. Is it "higher accuracy/better fitting," "reduced time requirements," "ease of implementation," "adoption of new technologies," or something else? Please provide more precise details regarding the aim of your study.
• In line 146: You can move the annotations to the figure (2) caption.
• In line 209, Figure 2: S1 to S10 I guess refer to Gas Sensors, so please refer to them in the figure caption. Later in the text (line 213) it is mentioned that “Figure 3 shows the gas response curves of 10 sensors.
• In line 299: the role of “bottleneck layer" is not explained in the text.
• In line 357: Remove “Therefore”.
• In line 375: The flowchart of Figure 7 can be improved. The information is given but the setup and representation needs to be reconstructed.
· In line 607: "There is no "conclusions" section. Please include a brief summary detailing the key conclusions and discussing the potential utility of the findings obtained through the proposed methodology."
Author Response
请参阅附件

Reviewer 2 Report
Comments and Suggestions for Authors
2. Materials and Methods.
1. No information is provided as to which method was used to determine the total nitrogen content of the soil.
2. What principle was used to select the sensors for the experiment?
3. Perhaps we should give the approximate composition of the cracking gases determined in the experiment?
2.2.3 Response experiment of soil nutrient detection device.
1. Please give more information about the concentration and proportion of gas in the total mixture, which can confirm the response to the presence of the stimulus and its quantification.
2. What does the term "foreign soil samples" mean?
Line 253: is the reference to Table 3 correct?
Line 260: here and further in the tables, symbols and terms must be deciphered
Lines 257-279: the meaning of this information and how this relates to Table 2 is not clear at all.
Line 375 - Figure. 7 Flow chart of the algorithm and Lines 481-482 - Figure 7. Feature signal diagram after pretreatment: (a) Features extracted by convolutional noise reduction autoencoder; ((b) Features extracted according to formula (1)- formula (4). Are the figure numbers repeated?
4. Discussion. Without denying the importance of the presented data processing model, I do not see a direct link between the analyses performed and the determination of total soil N content. The authors focused on the technical side of the issue with little attention to the practical application of the method and the reliability of the interpretation of the values. This section needs serious revision with more attention to soil nitrogen status, statistical methods, the reason for the choice of gas sensors and the practical usefulness of the results obtained.
Round 2
Reviewer 2 Report
Comments and Suggestions for Authors
all my questions have been answered